# Meta-Analysis on the Global Prevalence of Tetracycline Resistance in *Escherichia coli* Isolated from Beef Cattle

**DOI:** 10.3390/vetsci10070479

**Published:** 2023-07-21

**Authors:** Yohannes E. Messele, Gebremeskel Mamu Werid, Kiro Petrovski

**Affiliations:** 1Davies Livestock Research Centre, School of Animal & Veterinary Sciences, University of Adelaide, Roseworthy Campus, Roseworthy, SA 5371, Australia; yohannes.messele@adelaide.edu.au (Y.E.M.); gebremeskelmamu.werid@adelaide.edu.au (G.M.W.); 2Australian Centre for Antimicrobial Resistance Ecology, School of Animal & Veterinary Sciences, University of Adelaide, Roseworthy Campus, Roseworthy, SA 5371, Australia

**Keywords:** antimicrobial resistance, fecal samples, indicator organism, in-feed administration, treatment

## Abstract

**Simple Summary:**

Antimicrobial resistance (AMR) is a concern that impacts both human and animal health. To understand AMR, a detailed analysis of 14 different studies was carried out. This study specifically focused on resistance to tetracycline, a common antibiotic, in *E. coli* bacteria present in cattle. The study found that 0.31 of the *E. coli* from beef cattle not treated with antibiotics were resistant to tetracycline. Surprisingly, when beef cattle were given the antibiotic through feed or injection, the resistance only rose slightly to 0.53 and 0.39, respectively. This challenges the common belief that using lots of antibiotics in livestock causes higher resistance. The results varied greatly across studies, likely due to different factors like cattle genetics, environment, management, and how antibiotics are given. Other factors, such as exposure to antibiotics in the environment, natural resistance mechanisms in *E. coli*, and the ability of bacteria to share resistance traits, could also play a part. The study was limited by inconsistent data and a lack of standardization across the studies. Hence, the results highlight the need for tailored research on AMR, to gain a comprehensive understanding of the influencing factors and to devise effective countermeasures.

**Abstract:**

Antimicrobial resistance (AMR) is an emerging global concern, with the widespread use of antimicrobials in One Health contributing significantly to this phenomenon. Among various antimicrobials, tetracyclines are extensively used in the beef cattle industry, potentially contributing to the development of resistance in bacterial populations. This meta-analysis aimed to examine the association between tetracycline use in beef cattle and the development of tetracycline resistance in *Escherichia coli* isolates. A comprehensive search was conducted using multiple databases to gather relevant observational studies evaluating tetracycline use and tetracycline resistance in *Escherichia coli* isolates from beef cattle. The rate of tetracycline resistance from each study served as the effect measure and was pooled using a random-effects model, considering possible disparities among studies. The meta-analysis of 14 prospective longitudinal studies resulted in a 0.31 prevalence of tetracycline resistance in *Escherichia coli* in non-intervention (no exposure), contrasting numerically elevated resistance rates in the intervention (exposed) groups of 0.53 and 0.39 in those receiving tetracyclines via feed or systemically, respectively. Despite the observed numerical differences, no statistically significant differences existed between intervention and non-intervention groups, challenging the conventional belief that antimicrobial use in livestock inherently leads to increased AMR. The findings of this study underscore the need for additional research to fully understand the complex relationship between antimicrobial use and AMR development. A considerable degree of heterogeneity across studies, potentially driven by variations in study design and diverse presentation of results, indicates the intricate and complex nature of AMR development. Further research with standardized methodologies might help elucidate the relationship between tetracycline use and resistance in *Escherichia coli* isolated from beef cattle.

## 1. Introduction

The use of antimicrobials in animals plays a vital role in maintaining animal health and ensuring the safety and productivity of livestock industries. Antimicrobials are used mainly for the treatment and prevention of diseases [1]. The use of antimicrobials for prophylactic or preventative purposes is a common practice in various aspects of animal husbandry, especially in settings where the risk of infection is high. This is especially true in intensive farming environments such as beef feedlots where cattle are kept in close proximity to one another, creating conditions that can facilitate the spread of pathogens. Bovine Respiratory Disease Complex (BRDC) is one of the significant challenges for the beef industry, characterized by high morbidity and mortality rates, and high economic costs [2]. BRDC is a complex syndrome influenced by factors such as the health condition of the animal, environment, diet, transport, and immunity, posing a higher risk to cattle in feedlot systems [3]. Globally, the most common industry practice for preventing BRDC in high-risk cattle upon arrival at the feedlot is the metaphylactic use of antimicrobials [4,5,6]. The metaphylactic use of antimicrobials is employed to proactively control the potential development and spread of bovine respiratory disease (BRD) within a group of cattle, regardless of whether only a subset of animals exhibits respiratory disease symptoms [7,8,9]. To help prevent diseases, tetracycline is commonly used in food-producing animals and it can be administered to livestock using either as injections or mixed with feed [10]. The use of antimicrobials for the treatment or prevention of BRDC impacts the intended harmful bacteria but can also affect beneficial commensal bacteria residing within cattle.

About 70 to 90% of antimicrobials given to cattle are excreted intact or as metabolites in the urine and faeces [11]. It has been indicated that even small concentrations of antimicrobials have the potential to persist in the environment and contribute to the selection of AMR bacteria in the soil [12]. These bacteria can then spread through various pathways, including direct contact, consumption of contaminated food products, or environmental exposure [13]. Furthermore, historically in most developed countries, tetracyclines, have been utilized for purposes beyond their medicinal scope, acting as enhancers of feed efficiency and growth promoters in animals of agricultural importance [14]. Despite the advantages of antimicrobial use in food animals, there is increasing concern over the rise of antimicrobial-resistant bacteria, which potentially pose a hazard to animal and human health. The inappropriate use of antimicrobials creates a selective pressure that favours the survival and proliferation of resistant strains [15]. Nonetheless, it’s crucial to understand that the emergence of antimicrobial resistance is not solely attributable to inappropriate use, but rather it’s influenced by all kinds of usage, encompassing both appropriate and inappropriate applications [16]. This is because any exposure to antimicrobials can exert selective pressure on bacteria, leading them to develop resistance over time. Hence, it is of utmost importance to reduce the overall utilization of antimicrobials, while ensuring that any remaining use is strictly limited to situations where it is appropriate and necessary. Despite the recognized significance of this issue, there remains a knowledge gap regarding this risk. For instance, the magnitude of the risk and how it varies depending on the class of antimicrobial drug used, the duration of treatment, and the method of administration are not fully understood. This uncertainty highlights the need for further research in this area. To attain a comprehensive understanding of the existing situation, conducting a meta-analysis of studies carried out across diverse geographical locations stands out as one of the most effective approaches.

Hence, the aim of this meta-analysis was to examine the association between tetracyclines use in beef cattle and the prevalence of tetracycline resistance in *Escherichia coli* (*E. coli)* isolates. This meta-analysis is specifically focused on tetracycline, a commonly used class of antimicrobials, and *E. coli*, a well-studied and prevalent bacterium. Due to the wide distribution and relatively well-documented interactions with antimicrobials, offering a robust basis for our study, *E. coli* was chosen as an ‘indicator organism’. We anticipate that our work will offer a better understanding of how the use of tetracycline over time affects the resistance of *E. coli*, providing valuable insights that could inform future strategies for combating antimicrobial resistance (AMR). Furthermore, this study could potentially shed light on how different antimicrobial classes and treatment durations contribute to AMR, and ultimately guides more responsible and effective usage of these antimicrobials.

## 2. Materials and Methods

The Preferred Reporting Items for Systematic Reviews and Meta-Analyses (PRISMA) guidelines were followed in developing and implementing the research questions, search, and screening protocols, and in reporting the results [17].

### 2.1. Search Strategy

A comprehensive literature search was conducted across databases including PubMed, Scopus, and Web of Science using the advanced search option of each of these databases. For each database, the search term: “(tetracycline) AND (cattle OR beef OR feedlot OR fattening)”, with slight modification to fit the specific database advanced search formatting requirements, was used to search in the title, abstract and keywords of an article. The search period concluded on 22 April 2023. In addition to the systematic search of articles in the three databases, to ensure literature saturation, manual searches of the reference lists of eligible and relevant articles were also carried out.

### 2.2. Inclusion and Exclusion Criteria

All articles downloaded from each database were imported into Endnote, exported to Covidence (https://app.covidence.org/, accessed on 22 April 2023) and screened in two steps: first, title and abstract, and second, full text. Both screening steps were based on the developed inclusion and exclusion criteria (Table 1). All articles uploaded on the Covidence website were screened by two authors and screening conflicts were resolved by another, senior author.

Current literature on tetracycline resistance in feedlot cattle is limited, particularly for observational and case-control studies. Furthermore, inconsistencies in the presentation of results, despite the utilization of similar study designs, complicate the challenge of comparing and interpreting these studies. To mitigate these issues, our study incorporated observational data derived from feedlot cattle at both entry and exit points of feedlots. The data gathered at the point of feedlot exit was utilized as a control, facilitating comparison with other studies implementing case-control designs. These case-control studies typically employed interventions through the administration of various tetracycline analogues, delivered either via injection or incorporated into feed.

### 2.3. Data Extraction and Quality Assessment

Data extraction was conducted independently by two authors, using a pre-designed data extraction form. Extracted data included the first author’s name, publication year, study location, total sample size, number of tetracycline-resistant *E. coli* isolates, and other relevant study characteristics. The quality of eligible studies was independently assessed by two authors, YEM and GMW, using the Joanna Briggs Institute Qualitative Assessment and Review Instrument tool (JBI-QARI, available at https://jbi.global/critical-appraisal-tools, accessed 5 April 2023). Discrepancies were resolved through consensus or discussion with a third author, when necessary.

### 2.4. Outcomes and Variables

The primary outcome was the prevalence of tetracycline resistance in *E. coli* isolates derived from feedlot cattle. This was assessed based on observational data derived from both the entry and exit points of feedlots. Key variables included the type of tetracycline analogues used (tetracycline, oxytetracycline, chlortetracycline), the method of administration (injection or incorporated into feed), and the timing of sample collection (at entry or exit from feedlots).

### 2.5. Meta-Analysis

The meta-analysis was conducted using the ‘metafor’ package in R [18]. The proportion of tetracycline-resistant *E. coli,* in each study, was used as the effect measure for our binary outcomes. The proportion of tetracycline resistance was pooled using the random-effects model to account for potential heterogeneity among studies. Confidence in our effect measure was assessed using the random-effects model and confirmed by sensitivity analyses. Heterogeneity was assessed using the I^2 statistic and Q test. To explore potential sources of heterogeneity, Subgroup analyses were conducted based on predefined variables, such as intervention type and study design. Moreover, to assess the robustness of the findings, sensitivity analysis was conducted using the metafor package in R. The Graphic Display of Heterogeneity (GOSH) plots [19,20] were utilized to visually inspect the distribution of effect sizes and assess potential sources of heterogeneity. Additionally, an Influence analysis was conducted for each of the groups included in the study [21]. Given the nature of epidemiological data, which often exhibits noise and non-normal distributions, and due to its efficiency in handling these characteristics and its lack of reliance on assumptions of convex clusters or normal distributions, DBSCAN was selected for outlier detection [20]. Following the GOSH analysis, outliers identified by DBSCAN and those identified by Influence analysis were removed from the dataset. After removing these outliers, to ensure the integrity of the analysis, the meta-analysis was repeated. The potential presence of publication bias was assessed using Peter’s [22] and Egger’s [23] regression tests. In all analyses, a *p*-value less than 0.05 was considered to indicate statistical significance.

## 3. Results

### 3.1. Study Selection

The systematic literature search yielded a total of 6731 articles, of which 1108 were found in all three databases (Figure 1 and Figure 2). After removing 2786 duplicates, 3945 articles were screened for relevance based on title and abstract. This resulted in the exclusion of 3875 articles, leaving 70 full-text articles for assessment. Upon conducting a full-text screening, studies that failed to meet the pre-established inclusion and exclusion criteria, or did not pass the quality appraisal tool, were excluded from the final analysis. The reasons for their exclusion were as follows: incorrect study design (n = 36), non-comparable results (n = 16), inappropriate study population (n = 2), full text not accessible (n = 1), language other than English (n = 1), and unsuitable sample type (n = 1). In addition to these, an article was manually retrieved from Google Scholar, thus elevating the total number of articles included in the meta-analysis to 14 (Figure 1). 

From the 14 included studies, 4 were found exclusively in PubMed, 1 exclusively found in Scopus Seven of the 14 articles were found in all three databases (Figure 2).

### 3.2. Study Characteristics

The initial search protocol included both *Enterococcus* and *Salmonella* species. However, no eligible studies on these species were identified during the screening process. Consequently, our investigation was constrained to studies focusing solely on *E. coli*. In addition to analyzing minimum inhibitory concentration (MIC) data, we attempted to identify studies employing polymerase-chain reaction (PCR) or whole-genome sequencing (WGS) for the determination of tetracycline resistance. Despite our efforts, we were unable to locate studies employing these molecular methods for AMR determination that were comparable and could be included in the analysis.

Similarly, despite the search protocol encompassing articles published up until April 22, 2023, the studies ultimately included in our analysis were those published between 2005 and 2022. These studies originated from a diverse range of regions, namely Canada (n = 9), the United States (n = 2), Europe (n = 2), and Australia (n = 2). The sample sizes within these studies exhibited considerable variation, ranging from as few as 30 to as many as 3512 beef cattle per study. Cumulatively, our study included a total of 20,140 cattle.

### 3.3. Prevalence of Tetracycline Resistance in Escherichia coli Isolated from Beef Cattle without Intervention

The overall pooled prevalence of tetracycline resistance in *E. coli* isolated from beef cattle without intervention was 0.31 (95% Confidence Interval (CI): 0.17–0.48). Notably, there was no significant difference in the prevalence between the ‘Entry’ (0.32, 95% CI: 0.01–0.80) and ‘Control’ (0.31, 95% CI: 0.14–0.51) subgroups (Figure 3).

### 3.4. Prevalence of Tetracycline Resistance in Escherichia coli Isolated from Beef Cattle after Intervention

The pooled prevalence of tetracycline resistance in *E. coli* isolated from beef cattle that had been administered tetracycline in feed was slightly higher (0.53, 95% CI: 0.21–0.84) compared to those that had received tetracycline via injection (0.39, 95% CI: 0.00–1.00). Moreover, considering all interventions, the overall pooled prevalence of tetracycline resistance (0.49, 95% CI: 0.24–0.74) was higher than the prevalence observed in the control group (0.31, 95% CI: 0.01–0.80) (Figure 4). Despite the observed slight increase, the intervention did not have a significant effect on altering the prevalence of tetracycline resistance in *E. coli*.

### 3.5. Heterogeneity and Subgroup Analysis

Substantial heterogeneity was observed among the included studies in both the intervention and non-intervention groups (I^2^ = 100%, *p* < 0.01). A subgroup analysis was carried out to investigate potential sources of this heterogeneity. Within the intervention group, there was no significant difference (*p* > 0.05) between the sub-therapeutic administration of tetracycline through injection or feed supplementation. Similarly, there was no statistically significant difference (*p* > 0.05) between the subgroups within the nonintervention group. After removing outliers identified by sensitivity analysis, there was no significant difference (*p* > 0.05) between the intervention and nonintervention groups.

### 3.6. Sensitivity and Publication Bias Analysis

Results from the GOSH and Influence analyses were instrumental in identifying potential outliers, assessing the influence of individual studies on the overall effect estimate, and evaluating the stability of the pooled results. For the group without intervention, both GOSH and Influence analysis identified articles ‘Alexander et al., 2008′ [26] and ‘Benedict et al., 2015′ [32] as outliers. Following exclusion from the analysis, the overall pooled prevalence increased to 0.34 (95% CI: 0.18–0.53), with an I^2^ value of 99%, a reduction from the original estimates (Appendix A).

Similarly, in the intervention group, articles ‘Alexander et al., 2008′ [26] and ‘Lefebvre et al., 2006′ [34] were identified as outliers by both GOSH and Influence analysis. Removal of these two studies resulted in a revised overall pooled prevalence of 0.46 (95% CI: 0.26–0.67) and an I^2^ value of 98%, again higher than the original estimates (Appendix A).

A regression test for funnel plot asymmetry was conducted separately for the intervention and without intervention groups to assess potential publication bias within each. For the intervention group, the test was not statistically significant (*p* = 0.8029), suggesting no evidence of publication bias. The limit estimate as the standard error approached zero was 0.2079, with a 95% confidence interval of −0.6119 to 1.0276. This wide confidence interval, encompassing both negative and positive values, indicated uncertainty about the true effect size when the standard error was nearly zero. Conversely, in the non-intervention group, the test for funnel plot asymmetry similarly found no statistically significant evidence of publication bias (*p* = 0.9733). The limit estimate for this group was −0.2476 with a 95% confidence interval of −0.7334 to 0.2382. Once again, this wide confidence interval suggested uncertainty about the true effect size in studies with a low standard error. Taken together, these results suggested that, for both groups, the studies included in our meta-analysis were not significantly influenced by publication bias.

## 4. Discussion

The focus of this meta-analysis was to provide a comprehensive assessment of tetracycline resistance prevalence in *E. coli* populations isolated from beef cattle across a range of geographically diverse studies. The meta-analysis included 14 studies on the prevalence of tetracycline resistance of *E. coli* isolated from beef cattle. The overall pooled prevalence of resistance in non-intervention group *E. coli* was 0.31. In contrast, a higher prevalence was observed in intervention groups, where tetracycline was administered via feed (0.53) or injection (0.39). However, the increased prevalence of resistance in the intervention groups was not significantly different compared to the non-intervention groups. Substantial heterogeneity was observed among the included studies, yet the sources of this heterogeneity remained elusive as no significant differences were found upon subgroup analysis.

In addition to the scarcity of observational cohort and case-control studies on tetracycline resistance prevalence in *E. coli* populations isolated from beef cattle, this meta-analysis faced another challenge: variability in the reporting of results across studies. Inconsistencies in the presentation of results among available studies, even among those employing similar study designs, added to the difficulty of extracting, comparing, and interpreting these studies, resulting in the inclusion of only 14 articles in the current study. This may have decreased the total number of studies potentially suitable for inclusion in the meta-analysis, highlighting the need for standardized data reporting methods in AMR research. Standardization in reporting in the future will facilitate synergy in systematic reviews and meta-analyses studies.

In the current study, the prevalence of tetracycline resistance in *E. coli* isolated from non-intervention beef cattle was 0.31. Interestingly, the introduction of tetracycline as an intervention—either via feed or injection—elicited a moderate elevation in resistance prevalence to 0.53 in the group receiving tetracycline through feed and a modest elevation to 0.39 in those administered systemically. This increase in tetracycline resistance among intervention subgroups may be due to the selective pressure exerted by the antimicrobial, which favors the survival and propagation of resistant strains [38,39]. It is possible that the relatively higher level of resistance observed in the feed subgroup resulted from prolonged, gradual antimicrobial exposure, and subtherapeutic levels, thereby fostering an increase in resistance. However, additional factors, such as dosage, administration frequency, and biological heterogeneity among cattle, may contribute to this finding.

A subtle trend toward a higher prevalence of tetracycline resistance in intervention groups, particularly distinguished by the route of administration was observed. While we noted a trend towards a higher prevalence of tetracycline resistance in the intervention groups, this was not statistically significant. Therefore, it is important to interpret our results with caution. The absence of statistical significance may not necessarily negate the existence of an effect. Rather, it reveals the complexities and inherent difficulties associated with such types of analyses, such as the diversity in study designs, variation in sample sizes, wide confidence intervals, and high heterogeneity across our dataset, which may have potentially obscured a true effect if it exists. Interestingly, the consistent prevalence of tetracycline resistance in non-intervention groups, including both “entry” and “control” groups, points towards the influence of other factors apart from mere tetracycline exposure. This high baseline resistance could also have potentially impeded our ability to discern a statistically significant difference when compared with the intervention group. The lack of significant differences between the intervention and non-intervention groups contradicts the notion that the widespread use of antimicrobials in animals results in the development of AMR. Likewise, earlier studies reported no significant variations in the occurrence of AMR in isolates obtained from animals, regardless of whether they were exposed to antimicrobials or not [40,41,42]. Environmental exposure to tetracycline residues [43], innate resistance mechanisms in certain *E. coli* strains [44], and the ability of these bacteria to share resistance genes [45] may be major contributors to the higher prevalence of resistant *E. coli* populations in non-intervention groups. In addition, in the presence of multi-resistance genetic elements, the use of other antimicrobials may have inadvertently promoted tetracycline resistance [46]. Recent evidence also highlights the potential for bacteria to develop resistance when exposed to non-antibiotic compounds used in the agricultural food industry [47]. Moreover, previous studies indicate that the age and diet of cattle appeared to play a significant role in the acquisition and development of AMR commensal microflora, compared to the subtherapeutic use of antimicrobials [40].

Moreover, due to potential biases introduced by variations in study design, sample size, geographic location, and sampling methodology among the included studies, the global prevalence of resistance must be interpreted with caution. The wide 95% CI for this estimate (0.17–0.48) indicates a high degree of uncertainty, which likely reflects the substantial heterogeneity of the included studies. These findings highlight the complex and multifactorial nature of AMR development, necessitating additional exhaustive research. The observed higher heterogeneity across the studies included in this meta-analysis might be indicative of a multitude of interacting variables influencing the prevalence of tetracycline resistance in *E. coli* isolated from beef cattle. These could span from genetic and environmental factors to livestock management practices and variations in antimicrobial administration. Enhanced heterogeneity suggests that the outcomes of individual studies are not merely the result of sampling error, but are also influenced by these different factors, each contributing to the overall resistance prevalence [48]. The high heterogeneity, therefore, suggests a complex interplay of factors that is difficult to put into a single aggregated measure. This realization places a spotlight on the necessity for caution when interpreting the overall pooled resistance prevalence and the possibility of not adequately reflecting the specific contexts of individual studies. Consequently, it prompts a call for more focused, context-specific research that can better account for this inherent variability and offer a more nuanced understanding of tetracycline resistance in differing beef cattle populations.

The lack of studies employing advanced molecular resistance determination techniques, such as PCR or WGS, was one of the limitations of our investigation. In addition, the increased heterogeneity observed among the studies highlights the urgent need for standardization of methodologies and reporting practices within the field of AMR research. This substantial variation between studies has the potential to obscure nuanced interpretations and dilute specific findings, thereby potentially diminishing the significance of the aggregated results. Uniformity in methodological design and reporting will ensure more reliable comparability and enhance the efficacy of meta-analyses in elucidating clear AMR trends and patterns. This call to action for the harmonization of research emphasizes the significance of international collaboration and shared guidelines in the fight against AMR.

## 5. Conclusions

This meta-analysis has shed light on the magnitude of tetracycline resistance in *E. coli* isolated from beef cattle and highlighted the complex interplay of variables driving AMR in beef cattle. Regardless of whether beef cattle were directly exposed to tetracycline during the feeding period or not, the prevalence of tetracycline resistance among feedlot cattle was comparable. The notion that subtherapeutic use of antimicrobial in livestock leads to widespread resistance may be outdated. The current findings call for a thorough review and improvement of antimicrobial research to better understand what drives AMR in general and in beef cattle in particular. A deeper understanding of AMR mechanisms allows the development of strategies tailored to the beef industry’s specific challenges. Harmonization and standardization of studies reporting AMR are urgently required.

## Figures and Tables

**Figure 1 vetsci-10-00479-f001:**
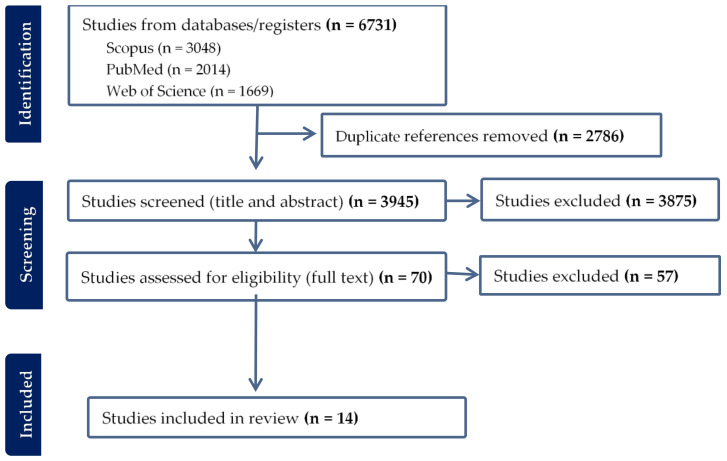
Article collection and screening steps followed and a few included studies.

**Figure 2 vetsci-10-00479-f002:**
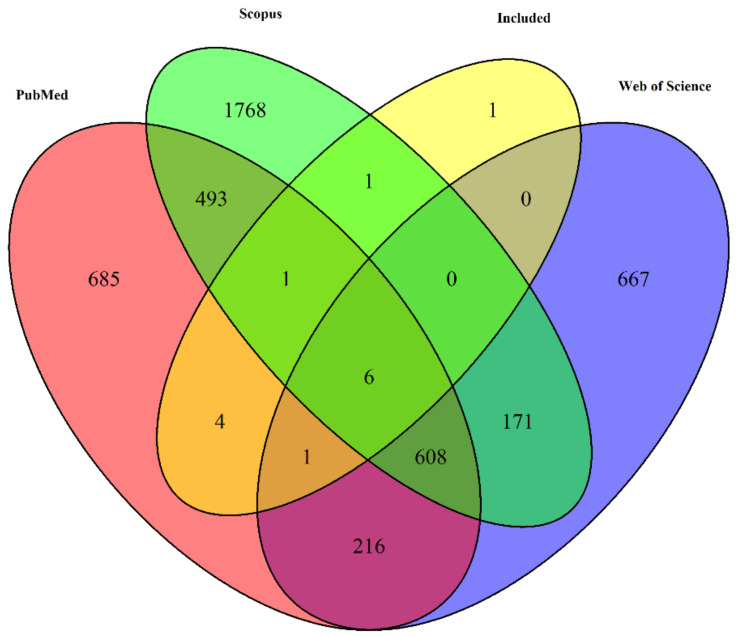
Venn diagram depicting the overlap of unique article titles retrieved from three different databases: PubMed, Web of Science, and Scopus, along with the articles that were included in the final selection.

**Figure 3 vetsci-10-00479-f003:**
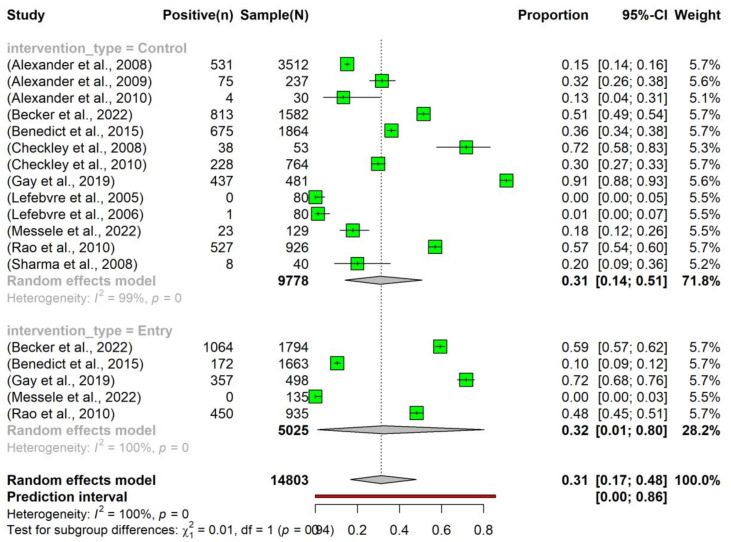
Prevalence of tetracycline resistance in *Escherichia coli* isolates obtained from beef cattle without antimicrobial intervention. Samples were collected upon the entry of cattle into the feedlots (‘Entry’) and at the time of their exit (‘Control’). NOTE: Some of the studies classified under the ‘Control’ subgroup were part of case-control studies (n = 7) [24,25,26,27,28,29,30], while others originated from cohort observational studies (n = 6) [31,32,33,34,35,36].

**Figure 4 vetsci-10-00479-f004:**
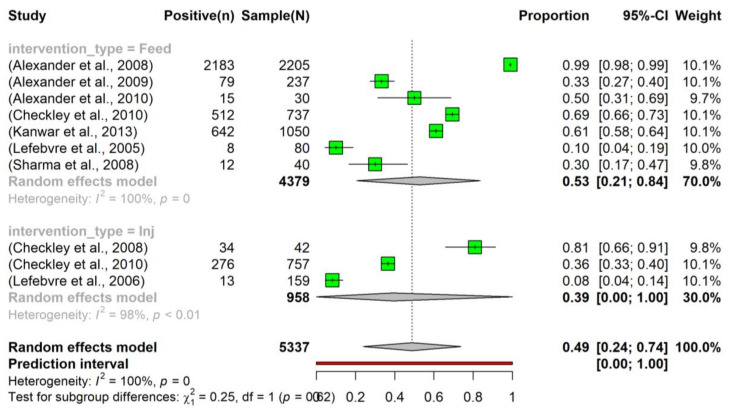
Prevalence of tetracycline resistance in *Escherichia coli* isolates sourced from beef cattle subjected to antimicrobial intervention. The tetracycline was administered sub-therapeutically via either feed (indicated as ‘Feed’; n = 7) [24,25,26,28,29,30,37] or an injection (denoted as ‘Inj’; n = 3) [27,28,34]. To examine the dynamics of tetracycline resistance following an intervention, samples were collected at various time points post-administration.

**Table 1 vetsci-10-00479-t001:** Inclusion and exclusion criteria.

Criteria	Inclusion Criteria	Exclusion Criteria
Study design	observational (cohort, case-control)	Reviews, editorials, commentaries, and non-observational studies (e.g., experimental, or interventional studies
Target bacteria	*Escherichia coli*, *Enterococcus* and *Salmonella*	Studies not specifically examining *Escherichia coli, Enterococcus* spp. or *Salmonella* spp.
Sample size	Greater than 30 samples	Less than 30 samples
Target animal	Beef cattle	Studies focused on dairy cattle, buffalo, or other species
Publication type	Peer review	non-peer-reviewed articles, conference abstracts, or unpublished data
Language	English	Non-English-language publications
Sample source	Fecal samples	Studies that use non-fecal samples, such as tissue, blood, or environmental samples.
Antimicrobial type	tetracycline, oxytetracycline, chlortetracycline	Studies using antimicrobials other than tetracyclines

## Data Availability

Data can be available upon request to authors.

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
