# Peer review of "Meta-Analysis on the Global Prevalence of Tetracycline Resistance in Escherichia coli Isolated from Beef Cattle"

_vetsci, 2023, doi:10.3390/vetsci10070479_

Round 1

Reviewer 1 Report

The study is well described and well conducted especially following PRISM framework.

Figure 1:  (n=???,

Figure 1: kindly state somewhere in the manuscript why you excluded 57, I know you have the inclusion and exclusion criteria but its important to still state why you excluded 57.

Line 197: Did you try metaregression in any way as part of the exploration for heterogeneity? Also, your subgroup analysis is limited to intervention vs control and feed vs inj, what about study design, year of publication, and any other metadata that account for the heterogeneity?

Author Response

Reviewer 1 comments and response from authors

Dear Reviewer,

We deeply value your insightful comments and constructive suggestions concerning our manuscript. For your convenience, we've provided our point-by-point responses below, each one labelled with 'AU' for easy reference, to address each comment or suggestion you've raised.

Reviewer 1 Point 1: Comments and Suggestions for Authors. The study is well described and well conducted especially following PRISM framework. Figure 1:  (n=???,

AU Response 1: Thank you for such a constructive suggestion. To make it clear, we have updated the total number of studies included within the figure and also indicated in the text (lines 194-195, indicated in a blue colour within the text)

Reviewer 1 Point 2: Figure 1: kindly state somewhere in the manuscript why you excluded 57, I know you have the inclusion and exclusion criteria but it’s important to still state why you excluded 57.

AU Response 2: Thank you for such a constructive suggestion. We have revised the result section of the manuscript (lines 188-195). To reflect these changes, we have also updated the methods section (lines 138-142, indicated in a blue colour within the text)

Reviewer 1 Point 3: Line 197: Did you try metaregression in any way as part of the exploration for heterogeneity? Also, your subgroup analysis is limited to intervention vs control and feed vs inj, what about study design, year of publication, and any other metadata that account for the heterogeneity?

AU Response 3: Yes, We conducted a meta-regression analysis incorporating variables such as country, study design, intervention type, and year (meta_reg <- metareg(tet_det_interv.prop, ~ Country + Design + intervention_type + year)). The results indicated that the combined influence of these covariates on the heterogeneity among studies did not reach statistical significance at the 0.05 level (p > 0.05). Thus, the variability in our study outcomes could not be adequately explained by these factors.

In our attempt to conduct a subgroup analysis, we faced several limitations due to the composition of our dataset. Specifically, the inclusion of only one type of study design - 'case-control' - restricted the feasibility of comparing outcomes across different study designs.

The geographical distribution of studies was also confined to only two countries, Canada and USA, hindering our ability to confidently interpret country-based variations. Subgroup analysis based on the country of origin could potentially yield skewed results due to the limited geographical representation.

Additionally, our efforts to incorporate the 'year of publication' as a subgroup variable proved challenging due to the uneven distribution of studies across years. The maximum number of studies published in a single year was just three, which could lead to unstable estimates and difficulties in drawing robust conclusions. Hence, we refrained from including the results of this line of analysis in this manuscript.

Reviewer 2 Report

This article presents a meta-analysis of tetracycline resistance in E. coli to assess the impact of treatment during the feeding period on the level of tetracycline resistance. E. coli was used as an indicator species, as it is ubiquitous in cattle, and tetracycline was selected due to its frequent use a preventative treatment for Bovine Respiratory Disease Complex (BRDC). It is well written.

The present study finds a moderate difference between the intervention and control populations, but with extremely broad error margins. Interestingly, they find a difference in average prevalence depending on the route of administration, with administration via feed producing higher levels of resistance (0.53, 95% CI: 0.21 - 0.84) relative to systemic administration (0.39, 95% CI: 0.00 - 1.00).

Removing outliers reduced both the the error margins 0.34 (95% CI: 0.18 - 0.53) -control-  vs 0.46 (95% CI: 0.26 - 0.67) -intervention group- but the difference between the two groups was still not statistically significant. The study concludes that  “the prevalence of tetracycline resistance among feedlot cattle was comparable”. I strongly disagree with this conclusion, the fact that the intervention group trended toward a higher prevalence of resistance and that the average prevalence depended on the route of administration, suggests the likelihood of an effect. In this case, the fact that the observed effect is not statistically significant is most likely due to a combination of insufficient sample size and to sample heterogeneity.  In any case, the lack of statistical significance is a negative result and therefore needs to be treated as such. Therefore I don’t think that  this study should be presented as evidence that exposure to tetracycline does not increase the prevalence of tetracycline resistance; I would just say that when different studies are combined the margin of error is so broad that it is not possible to draw a conclusion.

Having said that, the finding that in the non-intervention population the prevalence of resistance is around 0.31-0.32 is remarkably robust (found in both “entry” and “control” groups) and also biologically interesting, so my suggestion would be to build on this observation, as the authors begin to do in the discussion. This high background in the control group can be highlighted as a factor that reduces the ability to establish a statistically significant difference relative to the intervention group.  

I have also indicated in my entries that the design can be improved, by this I mean trying to increase sample size or to normalize the impact of each study by sample size if this has not been done already. 

Author Response

 Reviewer 2 comments and response from authors

Dear Reviewer,

We deeply value your insightful comments and constructive suggestions concerning our manuscript. For your convenience, we've provided our point-by-point responses below, each one labelled with 'AU' for easy reference, to address each comment or suggestion you've raised.

 Reviewer 2 Point 1: This article presents a meta-analysis of tetracycline resistance in E. coli to assess the impact of treatment during the feeding period on the level of tetracycline resistance. E. coli was used as an indicator species, as it is ubiquitous in cattle, and tetracycline was selected due to its frequent use a preventative treatment for Bovine Respiratory Disease Complex (BRDC). It is well written.

The present study finds a moderate difference between the intervention and control populations, but with extremely broad error margins. Interestingly, they find a difference in average prevalence depending on the route of administration, with administration via feed producing higher levels of resistance (0.53, 95% CI: 0.21 - 0.84) relative to systemic administration (0.39, 95% CI: 0.00 - 1.00).

Removing outliers reduced both the the error margins 0.34 (95% CI: 0.18 - 0.53) -control-  vs 0.46 (95% CI: 0.26 - 0.67) -intervention group- but the difference between the two groups was still not statistically significant. The study concludes that  “the prevalence of tetracycline resistance among feedlot cattle was comparable”. I strongly disagree with this conclusion, the fact that the intervention group trended toward a higher prevalence of resistance and that the average prevalence depended on the route of administration, suggests the likelihood of an effect. In this case, the fact that the observed effect is not statistically significant is most likely due to a combination of insufficient sample size and to sample heterogeneity.  In any case, the lack of statistical significance is a negative result and therefore needs to be treated as such. Therefore I don’t think that  this study should be presented as evidence that exposure to tetracycline does not increase the prevalence of tetracycline resistance; I would just say that when different studies are combined the margin of error is so broad that it is not possible to draw a conclusion.

AU Response 1: We agree that our meta-analysis provides some compelling insights and raises important questions, just as you pointed out. Regarding the non-significant difference between the intervention and control groups, our study intended to reflect the current state of evidence available in the literature. We agree that the trend toward higher prevalence in the intervention group, and the influence of the administration route (either by injection or feed), certainly suggests a possible effect. However, as the results did not reach statistical significance, we have chosen to present them as such to avoid overinterpretation.

We do understand your concern about the potential for sample size and heterogeneity to influence these results. Indeed, our analysis is reliant on the available studies, which vary considerably in their design and implementation, contributing to the heterogeneity we have observed.

 Reviewer 2 Point 2: Having said that, the finding that in the non-intervention population the prevalence of resistance is around 0.31-0.32 is remarkably robust (found in both “entry” and “control” groups) and also biologically interesting, so my suggestion would be to build on this observation, as the authors begin to do in the discussion. This high background in the control group can be highlighted as a factor that reduces the ability to establish a statistically significant difference relative to the intervention group.  

AU Response 2: We appreciate your suggestion to focus on the background prevalence of resistance in the non-intervention population, which indeed is a robust and biologically intriguing finding. We agree that this high baseline resistance level could potentially obscure the detection of a significant difference upon tetracycline exposure.

 Reviewer 2 Point 3: I have also indicated in my entries that the design can be improved, by this I mean trying to increase sample size or to normalize the impact of each study by sample size if this has not been done already. 

AU Response 3: We note that our work is a meta-analysis, and thus, we can only analyze the data already available in the literature. We do not have the ability to change the sample size or study design of the included studies. We did, however, use a random-effects model to account for the variability in study design and results.

Again, we are grateful for your feedback, and we will consider your points in our revision to provide a more nuanced interpretation of our results.

In response to the feedback, we have added an explanation in the discussion part of the manuscript (lines from  starting 317).

Reviewer 3 Report

I found this meta analysis as an important contribution to AMR. But I major changes should be done to your article in order to improve its quality.

Methodology:list the outcomes, variables, risk assessments (Specify the tool(s) (and version) used to assess risk of bias in the included studies), effect measures (Specify for each outcome (or type of outcome [e.g. binary, continuous]), the effect measure(s) (e.g. risk ratio, mean difference) used in the synthesis or presentation of results), explain the confidence assessment

Results: include table or figures for the risk of bias of the studies and certainty of evidence.

Best regards,

Author Response

Reviewer 3 comments and response from authors

Dear Reviewer,

We deeply value your insightful comments and constructive suggestions concerning our manuscript. For your convenience, we've provided our point-by-point responses below, each one labelled with 'AU' for easy reference, to address each comment or suggestion you've raised.

Reviewer 3 Point 1: I found this meta analysis as an important contribution to AMR. But I major changes should be done to your article in order to improve its quality.

Methodology: list the outcomes, variables, risk assessments (Specify the tool(s) (and version) used to assess risk of bias in the included studies), effect measures (Specify for each outcome (or type of outcome [e.g. binary, continuous]), the effect measure(s) (e.g. risk ratio, mean difference) used in the synthesis or presentation of results), explain the confidence assessment

AU Response 1: Thank you for your insightful comments. We appreciate the opportunity to provide more details on the outcomes, variables, risk assessments, effect measures, and confidence assessments used in our manuscript. To address these concerns, the methods section has been revised (lines starting from 146 indicated in blue font colour)

Reviewer 3 Point 2: Results: include tables or figures for the risk of bias of the studies and certainty of evidence.

AU Response 2: Thank you for your insightful comments. The results section has been updated (lines starting from 271, indicated in blue font colour). For more information, funnel plots have been added in the supplementary file.

Reviewer 4 Report

Review report

A brief summary: In this study the authors performed a meta-analysis to examine the association between tetracyclines use and the prevalence of tetracycline resistance in Escherichia coli in beef cattle.

Broad comments: The manuscript is well written, interesting, conclusions are in line with the results. I suggest some minor revisions in order to improve the manuscript and exclude any misinterpretations for the readers.

Specific comments 

Line 46-47: I should add also general health conditions of the animals and animal management at the farm as critical factors influencing the outcome of the syndrome.

Line 49: The authors should clarify if they speak about metaphylactic use or prophylactic or both of these kind of use.

Line 63: Antimicrobial resistance constitutes a threat also for animal health, please add.

Line 63-65: The selective pressure and the development of resistance is unfortunately not only caused by inappropriate use but by all use of antimicrobials (appropriate and inappropriate). This is why use of antimicrobials has to be reduced and only appropriate use can be allowed in order to slow down antimicrobial resistance. Unfortunately, we will never be able to stop selective pressure even when all inappropriate use is removed. Please reconsider the formulation of this sentence.  

Figure 1: I suggest to add in the figure also article number 14 which has been included in the meta-analysis. If not the reader may think you used only 13 articles.

Beef cattle without intervention: besides tetracyclines, were the animals of this group may be treated with other antimicrobials along their life ? Exposition to other antimicrobials may explain resistance to tetracyclines found in these animals. If the authors cannot exclude this possibility, I suggest to add this information.

Author Response

Reviewer 4 comments and response from authors

Dear Reviewer,

We deeply value your insightful comments and constructive suggestions concerning our manuscript. For your convenience, we've provided our point-by-point responses below, each one labelled with 'AU' for easy reference, to address each comment or suggestion you've raised.

Reviewer 4 Point 1: The manuscript is well written, interesting, conclusions are in line with the results. I suggest some minor revisions in order to improve the manuscript and exclude any misinterpretations for the readers.

AU Response 1: Thank you for your constructive comments.

Reviewer 4 Point 2: Line 46-47: I should add also general health conditions of the animals and animal management at the farm as critical factors influencing the outcome of the syndrome.

AU Response 2: We appreciate your provision of important information. We have now added more information from Line 63-65, indicated in blue font colour).

Reviewer 4 Point 3: Line 49: The authors should clarify if they speak about metaphylactic use or prophylactic or both of these kind of use.

AU Response 3: We appreciate your constructive comments. We have now added more information from Lines 67-69, indicated in blue font colour.

Reviewer 4 Point 4: Line 63: Antimicrobial resistance constitutes a threat also for animal health, please add.

AU Response 4: Thank you for your constructive comments. The importance of antimicrobial resistance in animals and humans is mentioned (lines 81-83, indicated in blue font colour).

Reviewer 4 Point 5: Line 63-65: The selective pressure and the development of resistance is unfortunately not only caused by inappropriate use but by all use of antimicrobials (appropriate and inappropriate). This is why use of antimicrobials has to be reduced and only appropriate use can be allowed in order to slow down antimicrobial resistance. Unfortunately, we will never be able to stop selective pressure even when all inappropriate use is removed. Please reconsider the formulation of this sentence. 

AU Response 5: Thank you for providing valuable information. We have now incorporated additional supporting details that reinforce the suggestion (lines 85-92, indicated in blue font colour).

Reviewer 4 Point 6: Figure 1: I suggest to add in the figure also article number 14 which has been included in the meta-analysis. If not the reader may think you used only 13 articles.

AU Response 6: Thank you for such a constructive suggestion. To make it clear, we have updated the total number of studies included within the figure and also indicated in the text (lines 194-195, indicated in a blue color within the text)

Reviewer 4 Point 7: Beef cattle without intervention: besides tetracyclines, were the animals of this group may be treated with other antimicrobials along their life ? Exposition to other antimicrobials may explain resistance to tetracyclines found in these animals. If the authors cannot exclude this possibility, I suggest to add this information.

AU Response 7: Thank you for your constructive comments. Yes, the high baseline resistance could also have impeded our ability to discern a statistically significant difference compared to the intervention group. Additionally, antimicrobial resistance is often associated with the selective pressure exerted by specific antimicrobials. However, it is crucial to acknowledge the phenomenon of co-selection for AMR, which involves the influence of other antimicrobials. Recent evidence also highlights the significant potential for bacteria to develop resistance when exposed to non-antibiotic compounds used in the agricultural food industry.

We have now incorporated additional information from Line 317-341

Round 2

Reviewer 2 Report

Thank you for incorporating two of my suggestions. My comment # 3 regarding the study design referred to filtering the data in a different way so as to possibly capture more studies. The large difference between number of initial hits versus the number of studies included suggests that the quality criteria used may have been too narrow, something to possibly keep in mind moving forward.

Reviewer 3 Report

Dear Sir,

I find the article relevant to the field. All the recommendations were done.